# Systematic review of the best evidence for resistance exercise in maintenance hemodialysis patients

**Qian Zhao[1☯], Ning Wu[1☯], Kaixing Duan[1,2☯], Jiahui Liu[1,2], Minghua Han[3], Huize Xu[3], Haoyang Chen[4]\*, Ji Ma[5]\***

1 Department of Nursing, Shanxi Provincial People's Hospital, Taiyuan, Shanxi, China, 2 School of Nursing, Shanxi University of Traditional Chinese Medicine, Jinzhong, Shanxi, China, 3 School of Nursing, Shanxi Medical University, Taiyuan, Shanxi, China, 4 Department of Nursing, The Rehabilitation Hospital of Nantong, Nantong, Jiangsu, China, 5 The Orthopaedic Spinal Ward, Shanxi Provincial People's Hospital, Taiyuan, Shanxi, China

☯ These authors contributed equally to this work.
\* chenhaoyangy@126.com (HC); Majisxty@163.com (JM)

**Data Availability Statement:** All relevant data are within the paper and its Supporting information files.

## Abstract

### Objective

This study aims to search, evaluate, and consolidate the best evidence for resistance exercise in maintenance hemodialysis patients, providing evidence-based support for the clinical implementation of resistance exercise in these patients.

### Methods

We conducted a comprehensive search of literature in databases on resistance exercise for maintenance hemodialysis patients, including guidelines, expert consensus, evidence summaries, systematic reviews, and randomized controlled trials. The search spanned from the inception of the database to March 2023. During the process of evaluation and data extraction, two researchers rigorously assessed the quality of the literature.

### Results

A total of 24 articles were included in this review, consisting of 2 guidelines, 3 expert consensus documents, 9 systematic reviews, and 10 randomized controlled trials. From nine aspects, including target population, contraindications for exercise, pre-exercise assessment, exercise frequency, exercise intensity, exercise duration, exercise type, exercise benefits, and exercise precautions, we extracted a total of 23 pieces of best evidence.

### Conclusion

Given the findings of this study, we recommend that future researchers design and conduct larger-scale, multi-center, longitudinal studies to validate our results and further explore the long-term impacts of combined resistance and aerobic exercises on muscle strength and

**Funding:** This study was financially supported by the 2022 Clinical Nursing Research Project of the Nursing Branch of the Chinese Society of Research Hospitals, titled "Study on the Promotion and Application of a Resistance Exercise Practice Programme for Patients with Stable Chronic Obstructive Pulmonary Disease in an Evidence-based Ecosystem", in the form of an award (Y2022FH-HLFH06-09) received by QZ. This study was also financially supported by the 2022-2023 Nursing Discipline Research Project of the Journal of the Chinese Medical Association, titled "Based on the Evidence Ecosystem", in the form of an award (CMAPH-NRD2022034) received by QZ. The funders had no role in study design, data collection and analysis, decision to publish, or preparation of the manuscript.

**Competing interests:** The authors have declared that no competing interests exist.

other health indicators. Such research will provide deeper insights and contribute to the development of evidence-based exercise programs.

## 1. Introduction

Maintenance hemodialysis (MHD) is a blood purification therapy and one of the crucial renal replacement therapies for patients with end-stage chronic kidney disease [1]. MHD can extend patients' lifespan and improve their quality of life, with its effectiveness second only to kidney transplantation [2]. Research indicates a growing global population of hemodialysis patients year by year [3], with China being the country with the highest number of individuals receiving hemodialysis [4]. By 2025, it is estimated that there will be 630 MHD patients per million population in China, with a total estimated number of hemodialysis patients reaching up to 870,000 individuals [5]. As the duration of dialysis extends, MHD patients experience varying degrees of cardiac and pulmonary function decline, muscle atrophy, physiological, psychological, and cognitive impairments, which significantly impact their quality of life [6]. Currently, both domestic and international research have provided a clear understanding of the mechanisms and intervention effects of exercise therapy for MHD patients. As a non-pharmacological intervention for MHD patients, exercise therapy can prevent muscle atrophy, improve physical function, and alleviate fatigue [7]. Specifically, when it comes to enhancing the muscle function of MHD patients, resistance exercise training has been shown to be more effective than aerobic exercise in promoting muscle gain [8], increasing muscle strength [9], enhancing upper limb grip strength [10], physical activity [11], improving systemic inflammatory responses [12], alleviating anxiety, depression, and other negative emotions, enhancing sleep quality, and improving overall quality of life [13,14]. The International Society of Renal Nutrition and Metabolism Global Kidney Exercise Team recommends increasing physical activity and exercise for all dialysis patients [15]. This study aims to comprehensively search for high-quality evidence literature regarding resistance exercise for MHD patients from both domestic and international sources. Through evaluation, data extraction, and summarization, we aim to provide a basis for healthcare professionals to develop scientific and rational resistance exercise programs for maintenance hemodialysis patients.

## 2. Materials and methods

### 2.1 Literature search

Utilizing the "6S" evidence model [16], a computerized search was conducted in the following databases and resources: BMJ Best Practice, UpToDate, Guidelines International Network (GIN), The National Institute for Health and Care Excellence (NICE), National Guideline Clearing House (NGC), Registered Nurses' Association of Ontario (RNAO), Scottish Intercollegiate Guidelines Network (SIGN), PubMed, Cochrane Library, web of science, China Medical Guideline Database, China National Knowledge Infrastructure (CNKI), Wanfang Database, VIP Database, and SinoMed Database, to retrieve all literature related to resistance exercise in maintenance hemodialysis(MHD) patients. English search terms included "hemodialysis/maintenance hemodialysis" and "exercise/strength training/resistance training/weight training," while Chinese search terms encompassed "血液透析/维持性血液透析" and "运动/力量训练/抗阻运动/负荷训练". The search was limited to publications from database inception until March 2023.

## 2.2 Inclusion and exclusion criteria for literature

Inclusion Criteria: Studies focusing on maintenance hemodialysis (MHD) patients as the study population; Research investigating resistance exercise in MHD patients; Types of studies eligible for inclusion: guidelines, expert consensus, evidence summaries, clinical decision-making studies, best practices, systematic reviews, and randomized controlled trials; Literature published in either Chinese or English. Exclusion Criteria: Literature types such as guideline interpretations, project proposals, and duplicate publications; Incomplete or inaccessible information within the documents; Studies that have not undergone quality assessment.

## 2.3 Criteria for literature quality assessment

The quality assessment for guidelines will be conducted using the Appraisal of Guidelines for Research and Evaluation (AGREE II) system [17]. Expert consensus will be evaluated according to the criteria established by the Joanna Briggs Institute (JBI) for Evidence-Based Healthcare (2016) [18]. Systematic reviews will undergo quality assessment using the Assessment of Multiple Systematic Reviews 2 (AMSTAR 2) [19]. Quality assessment for randomized controlled trials will adhere to the Joanna Briggs Institute's Randomized Controlled Trial Assessment Criteria (2016) [20]. We will employ the Australian JBI Evidence Recommendation Grading System (2014 version) to categorize recommendations into Grade A (strong recommendation) and Grade B (weak recommendation), following the FAME principles (Feasibility, Applicability, Clinical Significance, and Effectiveness) as the basis for recommendation levels [21].

## 2.4 Process of literature quality assessment

The quality assessment of literature will be conducted independently by two researchers who have undergone systematic evidence-based nursing training. In cases where these two researchers encounter discrepancies or conflicts during the assessment, a third party, who is an evidence-based nursing expert, will be consulted for resolution.

In instances where there is conflicting evidence or conclusions from different sources, this study will prioritize evidence that is systematic, of high quality, and most recently published [22].

## 3. Results

### 3.1 General characteristics of included literature

The literature screening flowchart and literature screening form in the S1 Fig. Flow chart and S5 Table. This study encompassed a total of 24 articles, which consisted of 2 guidelines [23,24], 3 expert consensus documents [25–27], 9 systematic reviews [9,28–35], and 10 randomized controlled trials [8,12,36–42]. Basic information regarding the included literature is provided in Table 1.

### 3.2 Results of literature quality assessment

**3.2.1 Quality assessment results of guidelines.** This study included two guidelines from Ashby and Baker [23,24]. The guidelines by Ashby et al. [21] scored standardized percentages of 100% in scope and purpose, 87.37% in stakeholder involvement, 70.65% in rigor, 85.32% in clarity, 68.32% in applicability, and 100% in editorial independence. The guidelines by Baker et al. [22] achieved standardized scores of 95.56% in scope and purpose, 68.42% in stakeholder involvement, 88.32% in rigor, 97.65% in clarity, 77.22% in applicability, and 100% in editorial independence. Both guidelines scored above 60% in all domains, resulting in an overall rating

**Table 1. Characteristics of included studies (n = 16).**

| Inclusion of literature | Literature sources | Type of study | Literature theme | Published (year) |
|---|---|---|---|---|
| Ashby et al. [23] | NICE | guidebook | Renal Association Clinical Practice Guidelines for Haemodialysis | 2019 |
| Baker et al. [24] | Medlive | guidebook | Exercise and lifestyle in chronic kidney disease | 2022 |
| Koufaki et al. [25] | PubMed | Expert consensus | Exercise therapy for patients with chronic kidney disease | 2015 |
| Renal Rehabilitation Professional Committee, Rehabilitation Physicians Branch, Chinese Medical Doctors' Association (CMA) [26] | China Knowledge Network (CNN) | Expert consensus | Expert consensus on exercise rehabilitation for adults with chronic kidney disease in China | 2019 |
| Wei Yuanyuan and others [27] | China Knowledge Network (CNN) | Expert consensus | Expert consensus on the construction of a renal rehabilitation system in haemodialysis units (centres) | 2021 |
| Gomes Neto et al. [28] | PubMed | Systematic Review | Effect of exercise training modalities in dialysis on physical function and health-related quality of life in maintenance haemodialysis patients | 2018 |
| Scapini et al. [29] | PubMed | Systematic Review | Aerobic, resistance and combined exercise improves exercise capacity and blood pressure control in maintenance haemodialysis patients | 2019 |
| Andrade et al. [30] | PubMed | Systematic Review | The effect of exercise on cardiorespiratory function in chronic kidney disease during dialysis | 2019 |
| Lu et al. [9] | PubMed | Systematic Review | Resistance training is effective in improving muscle mass and muscle strength in patients receiving dialysis | 2019 |
| Xu Qinjuan et al. [31] | Knowledge Network (CNN) | Systematic Review | The effect of different exercise modalities on improving walking ability in maintenance haemodialysis patients | 2021 |
| Cai et al. [32] | PubMed | Systematic Review | Analysis of the efficacy of aerobic exercise combined with resistance training in maintenance haemodialysis patients | 2022 |
| Dong et al. [36] | PubMed | Randomized controlled trial | Effect of resistance exercise in dialysis on systemic inflammation in maintenance haemodialysis patients with sarcopenia | 2019 |
| Zhang et al. [8] | PubMed | Randomized controlled trial | The effect of progressive resistance exercise in dialysis on the physical and quality of life of maintenance haemodialysis patients | 2020 |
| Zhao et al. [37] | PubMed | Randomized controlled trial | Aerobic exercise combined with resistance exercise improves dialysis adequacy and quality of life in maintenance haemodialysis patients | 2020 |
| Corrêa et al. [12] | PubMed | Randomized controlled trial | Effects of resistance training on sleep quality, redox balance and inflammatory status in maintenance haemodialysis patients | 2020 |
| Yansing et al. [38] | China Knowledge Network (CNN) | Randomized controlled trial | Effects of progressive resistance exercise on exercise capacity, nutritional indicators and sleep quality in maintenance haemodialysis patients | 2022 |
| Cheema B et al. [43] | PubMed | Randomized controlled trial | PEAK, a randomized controlled trial, investigates the effects of resistance training on anabolic metabolism in patients with kidney disease during their hemodialysis treatments | 2007 |
| de Lima et al. [39] | PubMed | Randomized controlled trial | Comparing the impacts of strength and aerobic exercises during hemodialysis sessions | 2013 |
| Matthew J. Clarkson et al. [34] | PubMed | Systematic Review | A systematic review and meta-analysis of exercise interventions aimed at enhancing physical function in end-stage kidney disease patients undergoing dialysis | 2019 |
| Sheng K et al. [33] | PubMed | Systematic Review | A meta-analysis of intradialytic exercise effects on hemodialysis patients | 2014 |
| Pellizzaro, Cíntia O et al. [40] | PubMed | Randomized controlled trial | The impact of peripheral and respiratory muscle training on the functional capabilities of patients undergoing hemodialysis | 2012 |
| Matsufuji, Shota et al. [41] | PubMed | Randomized controlled trial | A randomized controlled trial assessing the impact of chair stand exercise on the daily living activities of hemodialysis patients | 2015 |

(*Continued*)

**Table 1.** (Continued)

| Inclusion of literature | Literature sources | Type of study | Literature theme | Published (year) |
|---|---|---|---|---|
| Zhao QG et al. [35] | PubMed | Systematic Review | A systematic review of exercise interventions for individuals with end-stage renal disease | 2019 |
| Johansen, Kirsten L et al. [42] | PubMed | Randomized controlled trial | A study on the impact of resistance exercise training and nandrolone decanoate treatment on body composition and muscle functionality in hemodialysis patients | 2006 |

of Grade A. The literature, as a whole, exhibited high quality and was deemed suitable for inclusion (The results of the quality assessment of the guidelines are presented in the S1 Table).

**3.2.2 Quality assessment results of expert consensus.**　This study included two expert consensus documents. In document [25,26], all evaluation results were assessed as "yes" except for item 6, which was rated as "unclear." In document[27], the expert consensus had ratings of "yes" for all items except items 2 and 6, which were rated as "unclear." Both of the included documents exhibited complete research designs and overall good quality, qualifying them for inclusion (The results of the quality assessment results of expert consensus are presented in S2 Table).

**3.2.3 Quality assessment results of systematic reviews.**　This study included 9 systematic reviews, comprising one Chinese-language publication and five English-language publications. Among them, two systematic reviews, namely Lu et al. [9] and Cai et al. [32–35], received "yes" ratings for all assessment items, indicating comprehensive research design and high overall quality, and thus were eligible for inclusion. The systematic review by Gomes Neto et al. [28] received "yes" ratings for all items, except for item 15 ("Did the authors fully investigate publication bias?"), which was rated as "no." The research design was relatively complete, allowing inclusion. Scapini et al.'s systematic review [29] received "yes" ratings for all items except item 9 ("Did the authors use appropriate tools to assess the risk of bias in the included studies"), which received a "partially yes," and item 15 ("Did the authors fully investigate publication bias?"), rated as "no." The research design was relatively complete, justifying inclusion. Andrade et al.'s systematic review [30] received "yes" ratings for all items except item 10 ("Did the authors report the source of funding for the studies included in the systematic review"), which was rated as "no." The research design was relatively complete, permitting inclusion. Xu Qinjuan et al.'s systematic review [31] received "yes" ratings for all items except for item 10 ("Did the authors report the source of funding for the studies included in the systematic review"), and item 16 ("Did the authors report any potential conflicts of interest, including any funding received to conduct the systematic review?"), both of which were rated as "no." The overall quality was relatively high, justifying inclusion (The results of the quality assessment results of systematic reviews are presented in S3 Table).

**3.2.4 Quality assessment results of randomized controlled trials.**　This study included 10 randomized controlled trials, comprising 1 Chinese-language publication and 9 English-language publications. Among them, Dong et al.'s trial [36] and Zhao et al.'s trial [37,43] had ratings of "unclear" for items 4 ("Was blinding of participants implemented?") and 5 ("Was blinding of the interveners implemented?"), while all other item ratings were "yes." These trials demonstrated relatively complete designs, thus warranting inclusion. Zhang et al.'s trial [8,39] had a "unclear" rating for item 5 ("Was blinding of the interveners implemented?"), with "yes" ratings for all other items. The trial exhibited a relatively complete design, justifying inclusion. Corrêa et al.'s trial [12,40] had "not applicable" ratings for items 4 ("Was blinding of

participants implemented?") and 5 ("Was blinding of the interveners implemented?"), while all other items received "yes" ratings. The trial demonstrated a relatively complete design, supporting inclusion. Yan Xing's trial [38,41,42] received a "no" rating for items 4 ("Was blinding of participants implemented?") and 5 ("Was blinding of the interveners implemented?"), while all other item ratings were "yes." The trial exhibited a relatively complete design, justifying inclusion (The results of the quality assessment results of randomized controlled trials are presented in S4 Table).

### 3.3 Summary and description of evidence

Through the extraction and integration of evidence, this study has compiled evidence related to resistance exercise in maintenance hemodialysis (MHD) patients. Ultimately, 23 pieces of best evidence were extracted from nine aspects, including the target population for MHD patients' resistance exercise, contraindications to exercise, pre-exercise assessment, exercise frequency, exercise intensity, exercise duration, exercise type, exercise benefits, and exercise precautions, as detailed in Table 2.

## 4. Discussion

### 4.1 Comprehensive evidence summarization with clinical practice implications

In this study, evidence concerning resistance exercise in maintenance hemodialysis (MHD) patients was comprehensively summarized across nine key aspects: the appropriate population for MHD patients' resistance exercise, contraindications, pre-exercise assessment, exercise frequency, intensity, duration, type, benefits, and precautions, as outlined in Table 2. Evidence 1 highlights the appropriate population for resistance exercise among MHD patients, recommending exercise therapy for stable patients without contraindications [23,24]. Resistance exercise can enhance cardiorespiratory endurance, improve muscle strength and mass, reduce cardiovascular disease risk, alleviate emotional disturbances and sleep disorders, and enhance quality of life [26]. Evidence 2 to 6 detail contraindications for exercise. Prior to engaging MHD patients in resistance exercise, healthcare professionals should conduct a thorough assessment to identify and promptly address any contraindications to avoid potential health risks. Evidence 7 to 8 emphasize the importance of graded exercise testing (GXT) for MHD patients before undertaking moderate to high-intensity exercise. GXT evaluates patients' tolerance to progressive training, measures peak VO2, and facilitates the development of personalized training plans. Furthermore, it includes exercise testing to monitor blood pressure, blood oxygen levels, electrocardiograms, Borg's rating of perceived exercise (RPE), and clinical symptoms, ensuring the safety of patients during exercise testing [26]. In the safety of resistance in haemodialysis patients, no cases of cannula dislodgement during exercise were observed in any patient during the trial. Given that the majority of MHD patients struggle to complete maximal oxygen uptake (VO2max) assessments, alternative, simpler tests commonly used with other populations, such as the 6-minute walk test, are recommended for evaluating their functional capacity [27]. It is essential to note that MHD patients should undergo exercise testing on non-dialysis days and avoid measuring blood pressure on the side with a vascular access [26]. Evidence 9 to 14 describe the prescription of exercise for MHD patients based on the FITT principles. Engaging in appropriate rehabilitation exercise training during hospitalization is crucial for maintaining MHD patients' physical activity. However, outpatient rehabilitation exercise training is also indispensable. Healthcare professionals should provide patients and their families with relevant health education, with particular attention to assessing exercise

**Table 2. Summary of best evidence for resistance exercise in maintenance hemodialysis patients.**

| Form | Content of evidence | Recommended strength |
|---|---|---|
| applicable population | 1. It is recommended that all MHD patients without contraindications should engage in exercise during dialysis as a therapeutic way to increase physical function [23,24]; | A |
| Contraindications to exercise | 2. Abnormal blood pressure: severe hypertension (e.g., blood pressure over 180/110 mmHg), or hypotension (<90/60 mmHg) [26]; | A |
| | 3. Cardiopulmonary diseases: severe heart failure, arrhythmias, unstable angina, severe pericardial effusion, valvular stenosis, hypertrophic cardiomyopathy, aortic coarctation, etc., uncontrolled pulmonary hypertension (mean pulmonary artery pressure > 55 mmHg) [26]; | A |
| | 4. Acute clinical events: e.g. acute systemic inflammatory disease [26]; | A |
| | 5. Stop exercising immediately for symptoms of DVT such as unusual oedema, redness and pain in the calf [26]; | A |
| | 6. Those who can not cooperate with the exercise such as serious oedema, osteoarthrosis, etc. [26]; | A |
| Pre-exercise assessment | 7. It is recommended that MHD patients should undergo an exercise load test under the supervision of a healthcare professional prior to exercise to assess the patient's ability to tolerate incremental intensity exercise training [26]; | A |
| | 8. It is recommended that MHD patients under the supervision of healthcare personnel before exercise capacity test including cardiorespiratory endurance, muscular strength, muscular endurance, flexibility, etc., the test should be arranged on non-dialysis days; attention should be paid to avoid the measurement of blood pressure in the limb on the side of the inner fistula; commonly used methods of simple exercise capacity test are 6 min walking test, sit-to-stand test, and get-up-and-walk test to ensure the safety of the exercise process [26,27]; | A |
| | 9. A General Electric High Speed CT Scanner was used to perform CT scans on the nondominant mid-thigh of participants on a non dialysis day. These scans measured the thigh muscle cross-sectional area (CSA) and attenuation to assess muscle quantity and quality, respectively. Lower muscle attenuation values indicate better muscle quality due to less intramuscular lipid infiltration [44]. | A |
| | 10. the scans measured subcutaneous and total fat areas in the mid-thigh. All scans were collected and analyzed blindly, following previously reported methods [44]. | A |
| exercise frequency | 11. It is recommended that patients with MHD need to perform exercise training at least 3 times per week on top of increased daily physical activity [25,26]; | A |
| | 12. Exercise during routine haemodialysis was performed under the direct supervision of an exercise physiologist, with 8–15 sets of joint flexion and extension exercises 3 times per week for 12 weeks.[44] | A |
| exercise intensity | 13. Moderate-intensity aerobic exercise (50% to 70% VO2 peak) and resistance exercise (60% to 70% 1RM) are recommended for patients with MHD [25,26]; | A |
| | 14. It is recommended that patients with MHD need at least three to five sessions of low to moderate intensity exercise training per week [27]; | A |
| | 15. Peak force (kg) of the knee extensors, hip abductors, and triceps was measured bilaterally in triplicate with the best score recorded, using an isometric digital dynamometer (Chatillon CSD 200 Dynamometer; AMETEK, Paoli, PA; coefficient of variability 9.4%). These individual strength measures were summed to create a total strength measure [44]. | A |
| exercise duration | 16. The time of exercise was during or two hours before dialysis [33]; | A |
| | 17. The target duration is 30–60 min per exercise session, which can be divided into sessions depending on the individual MHD patient [26], the duration of exercise intervention should be longer than 6 month [34,39]; | A |
| Type of sport | 18. Suggested exercise patterns for people with MHD include aerobic exercise, resistance exercise, and flexibility training [27]; | A |
| | 19. Common resistance exercise programmes include: stretch pullers or elastic bandages, lifting dumbbells, sit-ups, push-ups, sand bags, leg weight, chair stand exercise etc. [26,40,41]; | A |
| | 20. Mainly low-to-moderate load lower extremity exercises with 1 to 3 sets of 8 to 15 repetitions per set [34]. | A |

(*Continued*)

**Table 2.** (Continued)

| Form | Content of evidence | Recommended strength |
|---|---|---|
| Exercise benefits | 21. Progressive resistance exercise in dialysis enhances musculoskeletal strength and exercise capacity in maintenance haemodialysis patients [25,26,28,29,31]; | A |
| | 22. Aerobic, resistance and combined exercise training improves dialysis adequacy and blood pressure control in maintenance haemodialysis patients [29]; | B |
| | 23. Resistance exercise in dialysis is effective in improving the systemic inflammatory response [36]; | B |
| | 24. Exercise programmes in dialysis can improve cardiorespiratory fitness, exercise tolerance and ventilatory efficiency in patients with chronic kidney disease [30]; | B |
| | 25. resistance training improves sleep quality, redox balance and inflammatory response in maintenance haemodialysis patients [12]; | B |
| | 26. Progressive resistance exercise enhances exercise capacity, improves nutritional status, and enhances sleep quality in MHD patients [38]; | B |
| | 27. Exercise can improve fatigue, anxiety, depression, physical activity, and QOL in patients with end-stage renal disease [35]; | A |
| | 28. Intradialytic exercise can increase the solute removal, for exercise may increase the blood flow to muscle, and greater toxic agents can be removed by the dialyzers [39]; | A |
| | 29. The exercise program of peripheral muscle resistance can increase the volume of anti-fatigue muscle fibers, the muscle's capture and transport of oxygen [33]; | A |
| Notes on exercise entry | 30. Exercise programmes should be individualised based on the patient's physiological function measurements and ability to perform daily activities, and it is recommended to start with low-intensity exercise training and gradually reach a moderate-intensity exercise level [26]; | A |
| | 31. MHD patients should immediately stop rehabilitation training if they experience: ① burning pain, soreness, and constriction in the chest, arms, neck, or jaw; ② severe chest tightness, shortness of breath, and dyspnea; ③ headache, dizziness, and general weakness; ④ severe cardiac arrhythmia; and ⑤ exercise-related muscle spasms and joint pains [26]; | A |
| | 32. Ensure that the safety of people with MHD is paramount and that safety precautions are in place before performing resistance exercise [25]. | A |
| | 33. The most common risk of intradialytic exercise is musculoskeletal injury, while the most serious risk is cardiovascular events, such as arrhythmia, myocardial infarction, and hypertension [39]; | A |
| | 34. Treatment with nanrodone caprate during weekly lower limb resistance exercise training is safe and well tolerated [42]; | A |

intensity and safety measures. Family members play a supervisory, supportive, and guiding role in the patient's exercise rehabilitation, encouraging them to gradually increase daily activity [26]. This should ensure the continuity and sustainability of exercise. Evidence 15 to 20 underscore the benefits of resistance training for MHD patients. Healthcare professionals should create individualized exercise prescriptions based on the patient's unique circumstances and preferences, encouraging them to remain consistent to reap health benefits. Evidence 21 to 23 outline exercise precautions. Due to the complex condition and numerous clinical comorbidities in dialysis patients, it is advisable for patients to start with low-intensity exercise training and gradually progress to moderate-intensity levels. Personalized prescriptions should be tailored according to the patient's physiological assessments and their activities of daily life (ADL) status. If there are indications for discontinuing exercise rehabilitation training, it should be halted immediately, prioritizing patient safety [26]. In line with the recommendations from the Exercise & Sport Science Australia (ESSA) position statement on exercise and chronic kidney disease (Smart NA, et al., 2013), our findings underscore the importance of a tailored approach to exercise prescription for individuals with ESKD. The ESSA guidelines emphasize the need for individualized assessment and the consideration of

patient-specific factors, which align with our study's approach to optimizing exercise training programs for this population.

Numerous studies have identified a variety of physical, psychological, and logistical barriers that impede the ability of hemodialysis patients to engage in regular exercise [45]. It is essential for nephrologists to address these barriers proactively, as highlighted by Clarke et al. [46], who emphasize the importance of nephrologists' special efforts in targeting patients' exercise barriers [47]. The proactive attitude of healthcare staff in dialysis centers is crucial for promoting a significant improvement in patients' levels of physical activity [48]. It is suggested that patient counseling and exercise prescription should be grounded in a multidisciplinary team-based approach. In this collaborative framework, the referring nephrologist should be supported by other healthcare professionals, such as physiotherapists and exercise physiologists, to ensure a comprehensive and personalized exercise program tailored to the unique needs of each patient. This multidisciplinary approach not only enhances the effectiveness of exercise interventions but also fosters a supportive environment that encourages patients to overcome the identified barriers and engage in regular physical activity, ultimately contributing to improved health outcomes.

To guarantee both safety and efficacy during the dialysis process, we advise against the use of the arm equipped with vascular access for strenuous resistance exercises. Instead, we suggest that exercise regimens be scheduled during the initial two hours of the dialysis session. This strategic timing is intended to circumvent the exhaustion that may arise from the increased net ultrafiltration volume typically observed in the final 1–2 hours of the dialysis treatment.

## 4.2 Evidence-based translation into clinical practice for resistance exercise in maintenance hemodialysis patients

While this study has provided a comprehensive summary of the best evidence for resistance exercise in maintenance hemodialysis (MHD) patients, it is essential to acknowledge that this evidence synthesis represents an integration of existing research findings. Current evidence suggests that resistance training had a positive impact on muscle strength, balance, and functional capacity in maintenance hemodialysis patients [43,49–52], though there is notable heterogeneity across studies due to variations in assessment indices, patient characteristics, and training protocols. Most studies report improvements in direct measures of muscle strength [51], while some also demonstrate gains in summary indices of strength and balance [53]. However, not all studies consistently show improvements in gait speed, STS test, or walking capacity as assessed by the 6-Minute Walk Distance (6MWD) [53]. These discrepancies highlight the need for standardized assessment methods and personalized training programs in future research to optimize the benefits and safety of resistance exercise for MHD patients. Ultimately, integrating resistance exercise into the daily routine of these patients holds promise for enhancing their overall quality of life. There may still be some discrepancies between the evidence summary and its practical implementation in clinical settings. Current randomized controlled trials examining resistance exercise in MHD patients exhibit certain limitations, such as small sample sizes and non-uniform intervention protocols. Additionally, several studies have investigated the combination of aerobic exercise with resistance exercise, further necessitating exploration and research into the singular effects of resistance exercise on MHD patients. When translating the evidence summary into practice, it is imperative to consider the individualized nature of clinical circumstances and the professional judgment of healthcare providers. A comprehensive and thorough assessment of patients, encompassing physical, psychological, social conditions, as well as the patients' preferences, is paramount. This holistic evaluation is essential for tailoring personalized and feasible resistance exercise regimens for

MHD patients, ensuring compliance, safety, and effectiveness. Future endeavors in the evidence-based translation of resistance exercise for MHD patients should prioritize the alignment of clinical realities, healthcare professionals' expertise, and a thorough patient evaluation. This approach guarantees the development of personalized and actionable resistance exercise plans that prioritize compliance, safety, and effectiveness.

### 4.3 Limitations

The heterogeneity in exercise training protocols, timing, and endpoints, as noted by the reviewer and highlighted in the work of Regolisti G et al. [38], introduces considerable variability in the outcomes reported across studies. This underscores the urgent need for a consensus on standardized protocols and endpoints to enhance the validity and generalizability of research findings in this field. Moreover, there is a clear necessity for standardized methods to assess physical function and HRQoL, which will provide a solid foundation for future comparative effectiveness studies.

## 5. Conclusion

Given the findings of this study, we recommend that future researchers design and conduct larger-scale, multi-center, longitudinal studies to validate our results and further explore the long-term impacts of combined resistance and aerobic exercises on muscle strength and other health indicators. Such research will provide deeper insights and contribute to the development of evidence-based exercise programs.

## Supporting information

**S1 Checklist. PRISMA 2020 checklist.**
(DOCX)

**S2 Checklist. PRISMA 2020 flow diagram for new systematic reviews which included searches of databases and registers only.**
(DOCX)

**S1 Fig. Flow chart.**
(DOCX)

**S1 Table. Quality assessment results of guidelines.**
(DOCX)

**S2 Table. Quality assessment results of expert consensus.**
(DOCX)

**S3 Table. Quality assessment results of systematic reviews.**
(DOCX)

**S4 Table. Quality assessment results of randomized controlled trials.**
(DOCX)

**S5 Table. All research information forms.**
(XLSX)

## Acknowledgments

We would like to thank Jian Zhao and Xiaoxiao Xue for their assistance with this study.

## Author Contributions

**Conceptualization:** Huize Xu.

**Data curation:** Qian Zhao.

**Formal analysis:** Ning Wu, Minghua Han, Ji Ma.

**Investigation:** Qian Zhao, Kaixing Duan, Huize Xu.

**Methodology:** Kaixing Duan, Huize Xu, Ji Ma.

**Project administration:** Ning Wu, Minghua Han.

**Resources:** Qian Zhao.

**Software:** Minghua Han, Haoyang Chen.

**Supervision:** Ning Wu, Kaixing Duan, Haoyang Chen.

**Validation:** Jiahui Liu, Haoyang Chen, Ji Ma.

**Visualization:** Ning Wu.

**Writing – original draft:** Qian Zhao, Kaixing Duan, Ji Ma.

**Writing – review & editing:** Kaixing Duan, Jiahui Liu.

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
