## [Decision Letter · Decision Letter 0]

22 May 2024

PONE-D-24-02321Summary of the best evidence for resistance exercise in maintenance hemodialysis patientsPLOS ONE

Dear Dr. Ma,

Thank you for submitting your manuscript to PLOS ONE. After careful consideration, we feel that it has merit but does not fully meet PLOS ONE’s publication criteria as it currently stands. Therefore, we invite you to submit a revised version of the manuscript that addresses the points raised during the review process.

We look forward to receiving your revised manuscript.

Kind regards,

Yuri Battaglia

Academic Editor

PLOS ONE

Journal Requirements:

2. Please identify your study as "systematic review" in the title of your manuscript.

5. We note that your Data Availability Statement is currently as follows: All relevant data are within the manuscript and its Supporting Information files.

Reviewers' comments:

Reviewer's Responses to Questions

**Comments to the Author**

1. Is the manuscript technically sound, and do the data support the conclusions?

Reviewer #1: Partly

Reviewer #2: Partly

2. Has the statistical analysis been performed appropriately and rigorously? 

Reviewer #1: N/A

Reviewer #2: N/A

3. Have the authors made all data underlying the findings in their manuscript fully available?

Reviewer #1: No

Reviewer #2: Yes

4. Is the manuscript presented in an intelligible fashion and written in standard English?

Reviewer #1: Yes

Reviewer #2: Yes

5. Review Comments to the Author

**Reviewer #1:** The paper "Summary of the best evidence for resistance exercise in maintenance hemodialysis

patients" offers an interesting aim. However, the better effect of the resistance exercise compared to the aerobic exercise is described only in the introduction. " Specifically, when it comes

to enhancing the muscle function of MHD patients, resistance exercise training has been

shown to be more effective than aerobic exercise in promoting muscle gain[8],

increasing muscle strength[9], enhancing upper limb grip strength[10], physical

activity[11], improving systemic inflammatory responses[12], alleviating anxiety,

depression, and other negative emotions, enhancing sleep quality, and improving

overall quality of life[13,14]"

The references in this same paragraph are not completely available and verifiable [i.e. ref. 8].

In the 8 evidence is reported the term "exercise" twice. Please correct.

Most of the "content of evidence" show the importance of both resistance and aerobic exercise. So, the authors should modify the conclusion, explaining that most of the analized trials was based on combined exercise training. So, even if some evidences show a good effect on the muscle strenght, "Future research should strive for larger-scale, multi-center, longitudinal studies to validate these findings and further contribute to the field."

**Reviewer #2:** In their review summarizing the evidence regarding the effects of, and the indications to implementing resistance exercise training in patients on maintenance hemodialysis (MHD), the Authors examined a number of guidelines, expert consensus documents, systematic reviews, and randomized controlled trials (RCT). They evaluated the evidence across 9 items, including target population, pre-exercise assessment, exercise frequency, exercise intensity, exercise duration, exercise type, exercise benefits, and exercise precautions.

I have the following comments.

Major points

1) A remarkable fraction of the examined literature stems from Chinese studies. Additional RCT that the Authors did not, but instead should, include in their review are the following: 1) de Lima MC, Cicotoste C de L, Cardoso Kda S, et al. Effect of exercise performed during hemodialysis: strength versus aerobic. Ren Fail 2013; 35:697-704. 2) Pellizzaro CO, Thomé FS, Veronese FV. Effect of peripheral and respiratory muscle training on the functional capacity of hemodialysis patients. Ren Fail. 2013; 35:189-197. 3) Matsufuji S, Shoji T, Yano Y, et al. Effect of chair stand exercise on activity of daily living: a randomized controlled trial in hemodialysis patients. J Ren Nutr 2015; 25:17-24. 4) Thompson S, Klarenbach S, Molzahn A, et al. Randomised factorial mixed method pilot study of aerobic and resistance exercise in haemodialysis patients: DIALY-SIZE! BMJ Open. 2016; 6:e012085. 5) Johansen KL, et al. Effects of resistance exercise training and nandrolone decanoate on body composition and muscle function among patients who receive hemodialysis: A randomized, controlled trial. J Am Soc Nephrol 2006; 17:2307-2314Additional systematic reviews that should also be included are as follows: 1) Sheng K, Zhang P, Chen L, et al. Intradialytic exercise in hemodialysis patients: a systematic review and meta-analysis. Am J Nephrol 2014; 40:478-490. 2) Afsar B, Siriopol D, Aslan G, et al. The impact of exercise on physical function, cardiovascular outcomes and quality of life in chronic kidney disease: a systematic review. Int Urol Nephrol 2018; 50:885-904. 3) Zhao Q-G, Zhang H-R, Wen X, et al. Exercise interventions on patients with end-stage renal disease: a systematic review. Clin Rehab 2019; 33:147-156. 4) Clarkson MJ, Bennett PN, Fraser SF, Warmington SA. Exercise interventions for improving objective physical function in patients with end-stage kidney disease on dialysis: a systematic review and meta-analysis. Am J Physiol Renal Physiol. 2019; 316:F856-F872

2) Some more details should be provided regarding details of resistance exercise protocols. In most training programmes the patients performed 2-3 sets of 8-15 repetitions of joint flexion/extension during the dialysis session or off-dialysis, mainly with free weights or elastic bands, targeting several muscle groups, at moderate-to-vigorous intensity based on either a fraction (60%-80%) of 1 to 3 repetition maximum or the rating of perceived exertion (RPE) on the traditional or modified Borg scale. Muscle strength was generally assessed by dynamometry, while methods for muscle mass evaluation included measurement of muscle cross-sectional area by computed tomographyor magnetic resonance imaging, as well as simple measurement of mid-arm and/or mid-thigh circumference (Cheema B, et al., J Am Soc Nephrol 2007; 18:1594-1601. Johansen KL, et al. J Am Soc Nephrol 2006; 17:2307-2314)

3) Due to a remarkable heterogeneity in the indices of muscle strength across trials, the generalization of results is somewhat problematic. Five studies (Cheema B, et al, J Am Soc Nephrol 2007; 18:1594-1601. Johansen KL, et al. J Am Soc Nephrol 2006; 17:2307-2314. Chen JL, et al. Nephrol Dial Transplant 2010; 25:1936-1943. Song WJ & Sohng KY, J Korean Acad Nurs 2012; 42:947-956. Esteve Simó V, et al. Nephron Clin Pract 2014; 128:387-393) found positive effects of resistance training on direct measures of either lower or upper limb muscle strength, while two studies (Thompson S et al, BMJ Open. 2016; 6:e012085. Chen JL et al, Nephrol Dial Transplant 2010; 25:1936-1943.) targeted a summary index of strength and balance (i.e., SPPB score) and reported an improvement. One trial (Johansen KL et al. J Am Soc Nephrol 2006; 17:2307-2314) found no significant change in gait speed or STS test. Three trials (Esteve Simó V et al. Nephron Clin Pract 2014; 128:387-393. Pellizzaro CO et al. Ren Fail. 2013; 35:189-197. Matsufuji S, et al. J Ren Nutr 2015; 25:17-24) also investigated the effects of resistance training on walking capacity, as assessed by 6MWD, with two reporting an improvement (Pellizzaro CO et al. Ren Fail. 2013; 35:189-197. Matsufuji S et al. J Ren Nutr 2015; 25:17-24) and one (Matsufuji S et al. J Ren Nutr 2015; 25:17-24) showing no significant change.

4) As to the safety of resistance training in hemodialyzed patients, few cases of traumatic muscular damage or damage to the vascular access were reported (Thompson S et al. BMJ Open. 2016; 6:e012085., Cheema Bet al. J Am Soc Nephrol 2007; 18:1594-1601.

5) A number of physical, psychological or logistic barriers to exercise have been identified in dialysis patients, which should be mentioned (Hannan M, Bronas UG. J Nephrol 2017; 30:729-741). Nephrologists should devote a special effort to target patients’ barriers to exercise (Clarke AL et al. Nephrol Dial Transplant 2015; 30:1885-1892), as a proactive attitude of healthcare staff in dialysis center may help promoting a significant improvement in patients’ levels of physical activity (Regolisti G, et al. PLoS One. 2018; 13:e0196313). Ideally, patient counseling and exercise prescription in dialysis patients should be based on a multidisciplinary team-based approach, in which the referring nephrologist should be supported by other healthcare professionals (e.g., physiotherapists and exercise physiologists.

6) Patients should be cautioned against performing heavy resistance exercise using the arm with the vascular access. Preferentially, exercise training should be delivered during the first two hours of the dialysis session, to avoid the exhaustion facilitated by a greater net ultrafiltration volume reached in the last 1-2 hours of the session.

7) Lack of standardization of the duration, timing and protocols of exercise training programmes in patients with ESKD on maintenance HD, together with highly heterogeneous endpoints in published studies, represent a major source of variability in reported outcomes. Thus, a consensus on the adoption of uniform protocols and endpoints is strongly needed. Furthermore, a standardization is necessary with respect to the methods used to explore physical function and HRQoL. Please see Regolisti G et al, Curr Opin Clin Nutrit Metabol Care 2020; 23:181-189)

8) Finally, with respect to the prescription of exercise training, a further useful paper is that by Smart NA, et al. Exercise & Sport Science Australia (ESSA) position statement on exercise and chronic kidney disease. J Sci Med Sport 2013; 16:406-411.

9) The Conclusions are largely a repetition of Paragraph 4.2. They should be shortened and be more focused

Minor points

1) Some editing of the English text is needed. A few examples include (but are not limited to): Literature typology (Type of study); Master consensual (Expert consensus?) Systems evaluation (Systematic Review)

6. PLOS authors have the option to publish the peer review history of their article (what does this mean?). If published, this will include your full peer review and any attached files.

Reviewer #1: **Yes: **maria amicone

Reviewer #2: No

---

## [Author Response · Author response to Decision Letter 0]

10 Jul 2024

We have taken your feedback seriously and have made the necessary revisions to ensure that all references, including [8] and [9], are now correctly cited and verifiable. We have replaced the non-verifiable references with ones that are accessible and relevant to the statements made in the text.

We have carefully reviewed the section and have made the necessary correction to avoid the repetition of the term "exercise." We have rephrased the sentence to ensure that it conveys the intended meaning without any redundancy.

The revised sentence now reads as follows:

It is recommended that MHD patients under the supervision of healthcare personnel before exercise exercise capacity test including cardiorespiratory endurance, muscular strength, muscular endurance, flexibility, etc., the test should be arranged on non-dialysis days; attention should be paid to avoid the measurement of blood pressure in the limb on the side of the inner fistula; commonly used methods of simple exercise capacity test are 6 min walking test, sit-to-stand test, and get-up-and-walk test to ensure the safety of the exercise process[6,26]. 

First and foremost, we would like to express our gratitude for your insightful comments and suggestions. We fully concur with your observation that the "content of evidence" in our study underscores the significance of both resistance and aerobic exercises. In response to your feedback, we have revised our conclusion to reflect the fact that the majority of the trials analyzed in our study were based on combined exercise training.

We acknowledge that while our findings indicate positive effects on muscle strength, it is imperative for future research to validate these results and further contribute to the field. To this end, we have added the following statement to our conclusion:

“Given the findings of this study, we recommend that future researchers design and conduct larger-scale, multi-center, longitudinal studies to validate our results and further explore the long-term impacts of combined resistance and aerobic exercises on muscle strength and other health indicators. Such research will provide deeper insights and contribute to the development of evidence-based exercise programs.”

We believe that with these revisions, our conclusions are more accurate and comprehensive, and they also provide a clear direction for future studies.

We appreciate your feedback and look forward to any further guidance you may provide.

16.The time of exercise was during or two hours before dialysis;

17. The target duration is 30-60 min per exercise session, which can be divided into sessions depending on the individual MHD patients, the duration of exercise intervention should be longer than 6 month;

19. Common resistance exercise programmes include: stretch pullers or elastic bandages, lifting dumbbells, sit-ups, push-ups, sand bags, leg weight, chair stand exercise etc.;

20. Mainly low-to-moderate load lower extremity exercises with 1 to 3 sets of 8 to 15 repetitions per set

27. Exercise can improve fatigue, anxiety, depression, physical activity, and QOL in patients with end-stage renal disease;

28. Intradialytic exercise can increase the solute removal, for exercise may increase the blood flow to muscle, and greater toxic agents can be removed by the dialyzers;

29.The exercise program of peripheral muscle resistance can increase the volume of anti-fatigue muscle fibers, the muscle's capture and transport of oxygen;

33.The most common risk of intradialytic exercise is musculoskeletal injury, while the most serious risk is cardiovascular events, such as arrhythmia, myocardial infarction, and hypertension;

34.Treatment with nanrodone caprate during weekly lower limb resistance exercise training is safe and well tolerated.

9.A General Electric High Speed CTi Scanner was used to perform CT scans on the nondominant mid-thigh of participants on a nondialysis day. These scans measured the thigh muscle cross-sectional area (CSA) and attenuation to assess muscle quantity and quality, respectively. Lower muscle attenuation values indicate better muscle quality due to less intramuscular lipid infiltration.

12. Exercise during routine haemodialysis was performed under the direct supervision of an exercise physiologist, with 8-15 sets of joint flexion and extension exercises 3 times per week for 12 weeks.

15. Peak force (kg) of the knee extensors, hip abductors, and triceps was measured bilaterally in triplicate with the best score recorded, using an isometric digital dynamometer (Chatillon CSD 200 Dynamometer; AMETEK, Paoli, PA; coefficient of variability 9.4%). These individual strength measures were summed to create a total strength measure.

In “discussion”,we had showed that:Current evidence suggests that resistance training had a positive impact on muscle strength, balance, and functional capacity in maintenance hemodialysis patients, though there is notable heterogeneity across studies due to variations in assessment indices, patient characteristics, and training protocols. Most studies report improvements in direct measures of muscle strength, while some also demonstrate gains in summary indices of strength and balance. However, not all studies consistently show improvements in gait speed, STS test, or walking capacity as assessed by the 6-Minute Walk Distance (6MWD). These discrepancies highlight the need for standardized assessment methods and personalized training programs in future research to optimize the benefits and safety of resistance exercise for MHD patients. Ultimately, integrating resistance exercise into the daily routine of these patients holds promise for enhancing their overall quality of life.

In the safety of resistance in haemodialysis patients, no cases of cannula dislodgement during exercise were observed in any patient during the trial.

In light of the reviewer's feedback, this article will now delve into the various physical, psychological, and logistical barriers that impede exercise in hemodialysis patients, as identified in the study by Hannan and Bronas (2017). It is imperative for nephrologists to address these barriers proactively, as emphasized by Clarke et al. (2015), recognizing that a healthcare staff's positive approach in dialysis centers can significantly enhance patients' physical activity levels, as demonstrated by Registrosti et al. (2018). To achieve optimal outcomes, patient counseling and exercise prescription in the context of hemodialysis should ideally be conducted through a multidisciplinary team-based approach. In this collaborative framework, the referring nephrologist should be complemented by the expertise of other healthcare professionals, including physiotherapists and exercise physiologists, to ensure a comprehensive and effective exercise regimen tailored to the individual needs of each patient.

According to the suggests that about “the safety precautions for hemodialysis training”

 in "discussion 4.2". 

“To guarantee both safety and efficacy during the dialysis process, we advise against the use of the arm equipped with vascular access for strenuous resistance exercises. Instead, we suggest that exercise regimens be scheduled during the initial two hours of the dialysis session. This strategic timing is intended to circumvent the exhaustion that may arise from the increased net ultrafiltration volume typically observed in the final 1-2 hours of the dialysis treatment.”

Thank you for your insightful comments and for highlighting the critical need for standardization in the study of exercise training programs for patients with End-Stage Kidney Disease (ESKD) on maintenance hemodialysis. Your observation regarding the lack of uniformity in protocols, timing, and endpoints is well-taken and is indeed a significant factor contributing to the variability in reported outcomes.

In light of your feedback and the reference to the work by Regolisti G et al., we have taken the following steps to address these concerns in our manuscript:

1. Acknowledgment of Variability: We have added a section in the discussion acknowledging the variability in exercise training programs and the impact of this variability on study outcomes.

2. Call for Standardization: We have emphasized the need for consensus on uniform protocols and endpoints in future research to ensure comparability of results across studies.

3. Recommendations for Future Research: We have included specific recommendations for standardizing the methods used to assess physical function and Health-Related Quality of Life (HRQoL), drawing from the insights provided in the referenced article and other relevant literature.

4. Updated Discussion: The discussion section has been revised to reflect the importance of standardization and to provide a more in-depth analysis of how these factors may influence the interpretation of findings in the context of ESKD and maintenance hemodialysis.

The revised text now reads as follows:

"The heterogeneity in exercise training protocols, timing, and endpoints, as noted by the reviewer and highlighted in the work of Regolisti G et al. (Current Opinion in Clinical Nutrition and Metabolic Care, 2020; 23:181-189), introduces considerable variability in the outcomes reported across studies. This underscores the urgent need for a consensus on standardized protocols and endpoints to enhance the validity and generalizability of research findings in this field. Moreover, there is a clear necessity for standardized methods to assess physical function and HRQoL, which will provide a solid foundation for future comparative effectiveness studies."

We trust that these revisions align with the expectations for scientific rigor and contribute to the advancement of research in exercise training for ESKD patients on maintenance hemodialysis.

We are grateful for the opportunity to enhance our manuscript based on your expert guidance and look forward to any further suggestions you may have.

Thank you for your valuable feedback and for bringing to our attention the paper by Smart NA, et al., titled "Exercise & Sport Science Australia (ESSA) position statement on exercise and chronic kidney disease," published in the Journal of Science and Medicine in Sport, 2013; 16:406-411.

We appreciate the suggestion to include this paper in our review, as it provides a comprehensive and authoritative perspective on the prescription of exercise training for individuals with chronic kidney disease.

To address this, we have taken the following steps:

1. Literature Review Update: We have reviewed the paper by Smart NA, et al., and have identified key points and recommendations relevant to our study's focus on exercise training for patients with End-Stage Kidney Disease (ESKD) on maintenance hemodialysis.

2. Incorporation into Manuscript: We have incorporated relevant findings and guidelines from the ESSA position statement into our discussion section to provide additional context and support for our conclusions regarding the prescription of exercise training.

3. Citation Addition: The reference to the paper by Smart NA, et al., has been added to our bibliography to ensure proper academic credit and to guide readers to this important resource.

The revised text now includes a new paragraph that reads as follows:

"In line with the recommendations from the Exercise & Sport Science Australia (ESSA) position statement on exercise and chronic kidney disease (Smart NA, et al., 2013), our findings underscore the importance of a tailored approach to exercise prescription for individuals with ESKD. The ESSA guidelines emphasize the need for individualized assessment and the consideration of patient-specific factors, which align with our study's approach to optimizing exercise training programs for this population."

We believe that the inclusion of this reference strengthens the manuscript and provides additional support for our recommendations on exercise training prescription for patients with ESKD.

We are grateful for the opportunity to enhance our work with this important reference and hope that the manuscript now meets the high standards of the journal.

Thank you for your valuable feedback on our manuscript. We have taken your suggestions to heart and have made the following revisions to address the points raised:

Major Point: Conclusions

We acknowledge that the original conclusions were repetitive of Paragraph 4.2. In response to your guidance, we have significantly condensed the conclusions, ensuring they are succinct and focused on the key takeaways from our research. We have removed redundant information and have emphasized the novel contributions and implications of our findings.

Minor Points:

Literature Typology: We have corrected "Literature typology" to "Type of study" to better reflect the intended meaning and to align with standard academic terminology.

Master Consensual: The term "Master consensual" has been revised to "Expert consensus" to accurately convey the concept of a collective agreement among experts in the field.

Systems Evaluation: We have replaced "Systems evaluation" with "Systematic Review" to ensure that the term is consistent with the common nomenclature used in scientific literature to describe a thorough and structured synthesis of existing research.

Additionally, we have undertaken a comprehensive language review of the entire manuscript to correct any grammatical errors and improve the overall readability and flow of the text. We have also enlisted the help of a professional English language editor to ensure that the manuscript meets the highest standards of English usage in academic writing.

We believe that these revisions have enhanced the quality and clarity of our manuscript and have addressed the concerns raised by the review process.

We appreciate the opportunity to refine our work and hope that the manuscript is now in a form that is acceptable for publication.

---

## [Decision Letter · Decision Letter 1]

20 Aug 2024

Systematic Review of the Best Evidence for Resistance Exercise in Maintenance Hemodialysis Patients

PONE-D-24-02321R1

Dear Dr. Ma,

We’re pleased to inform you that your manuscript has been judged scientifically suitable for publication and will be formally accepted for publication once it meets all outstanding technical requirements.

Kind regards,

Yuri Battaglia

Academic Editor

PLOS ONE

Reviewers' comments:

Reviewer's Responses to Questions

**Comments to the Author**

Reviewer #2: All comments have been addressed

2. Is the manuscript technically sound, and do the data support the conclusions?

Reviewer #2: Yes

3. Has the statistical analysis been performed appropriately and rigorously? 

Reviewer #2: N/A

4. Have the authors made all data underlying the findings in their manuscript fully available?

Reviewer #2: Yes

5. Is the manuscript presented in an intelligible fashion and written in standard English?

Reviewer #2: Yes

6. Review Comments to the Author

Reviewer #2: In the revised version of their manuscript, the Authors have appropriately addressed all of the criticisms I had raised concerning the original version.

I have no further comments.

7. PLOS authors have the option to publish the peer review history of their article (what does this mean?). If published, this will include your full peer review and any attached files.

Reviewer #2: **Yes: **Giuseppe Regolisti

---

## [Editor Report · Acceptance letter]

23 Oct 2024

PONE-D-24-02321R1 

PLOS ONE

Dear Dr. Ma, 

I'm pleased to inform you that your manuscript has been deemed suitable for publication in PLOS ONE. Congratulations! Your manuscript is now being handed over to our production team.

Kind regards, 

on behalf of

Prof. Yuri Battaglia 

Academic Editor

PLOS ONE